# PIKACHU: Prototypical In-context Knowledge Adaptation for Clinical Heterogeneous Usage

**Amar Kumar**[1,2]                                          AMAR.KUMAR@MAIL.MCGILL.CA

**Zahra TehraniNasab**[1,2]                          ZAHRA.TEHRANINASAB@MAIL.MCGILL.CA

**Emily Kaczmarek**[1,2]                               EMILY.KACZMAREK@MAIL.MCGILL.CA

**Tal Arbel**[1,2]                                              TAL.ARBEL@MCGILL.CA

[1] *Center for Intelligent Machines, McGill University, Montreal, Canada.*

[2] *Mila - Quebec AI Institute, Montreal, Canada.*

**Editors:** Accepted for publication on at MIDL 2026

## Abstract

Medical imaging systems increasingly rely on large vision language foundation models (VLFMs) trained on diverse biomedical corpora, yet these models remain difficult to adapt to new clinical tasks without costly fine-tuning and large annotated datasets. We present *PIKACHU* (*Prototypical In-Context Knowledge Adaptation for Clinical Heterogeneous Usage*), a lightweight and generalizable framework that enables rapid few-shot adaptation of frozen medical FMs using only a handful of labelled examples. Unlike prior approaches that modify backbone weights or introduce heavy attention-based adapters, *PIKACHU* performs all task adaptation directly in the FM feature space through *in-context prototypical reasoning*. Given a small support set, the framework constructs class prototypes by averaging normalized embeddings from a frozen VLFM image encoder and performs prediction on query images using temperature-scaled cosine similarity. Only a single temperature parameter is learned. We evaluate *PIKACHU* across three heterogeneous medical imaging datasets - dermatological images (ISIC), Optical Coherence Tomography (OCT), and Diabetic Retinopathy (DR), using established vision models (SigLIP, PubMedCLIP, DinoV2, and ViT) as backbones. The proposed in-context learning (ICL) strategy consistently outperforms the baseline (zero-shot) approaches across all datasets and architectures, achieving substantial improvements in both accuracy and AUC. Notably, with PubMed-CLIP as the backbone, *PIKACHU* achieves 0.69 accuracy on the ISIC dataset, 0.72 on OCT, and 0.79 on DR, demonstrating robust generalization across diverse clinical imaging modalities. These results highlight the promise of feature-space in-context learning as an efficient and deployable paradigm for test-time adaptation of foundation models, without the need for extensive retraining. To facilitate broader adoption and research, we make our code publicly available at https://github.com/Amarkr1/pikachu.

**Keywords:** Foundation Models, In-Context Learning, Large Language Models.

## 1. Introduction

Medical imaging is central to clinical diagnosis and treatment across a wide range of diseases and healthcare environments, leading to remarkable heterogeneity in real-world imaging data. For example, chest radiographs acquired across community clinics and tertiary-care hospitals often exhibit substantial differences in disease appearance, image quality, and acquisition protocols (Irvin et al., 2019; Johnson et al., 2019). Dermatology images captured

using consumer-grade smartphones show large variations in lighting, color balance, and zoom level (Tschandl et al., 2018), while MRI scans vary widely due to changes in echo time, slice thickness, and scanner field strength (Zhou et al., 2019). Despite this variability, most deep learning models are optimized for narrow, predefined tasks, such as classifying thoracic diseases in CheXpert (Irvin et al., 2019) or detecting retinal fluid in OCT (Kermany et al., 2018). When deployed in new settings involving previously unseen diseases (e.g., COVID-19 pneumonia) or changes in imaging acquisition protocol, model accuracy often degrades sharply due to out-of-distribution shifts (Cohen et al., 2020). Maintaining reliable performance across such shifts typically requires costly retraining and large new annotated datasets, which are rarely available in emerging or resource-constrained clinical environments.

Recently, self-supervised learning has enabled the training of large-scale foundation models (FMs) by deriving training labels directly from the data itself (Eslami et al., 2023; Oquab et al., 2023; Zhai et al., 2023). These models can therefore leverage data from numerous sources and diverse diseases to learn generalizable representations applicable across a range of downstream tasks. After pre-training, such downstream prediction tasks (e.g., classification) can be performed through zero-shot inference in vision-language FMs, or by training simple classifiers on top of frozen or fine-tuned vision FMs (such as Tip-Adapter (Zhang et al., 2022), Proto-Adapter (Kato et al., 2024), and LoRA (Hu et al., 2022)). While these strategies have shown success in numerous applications (Radford et al., 2021; Oquab et al., 2023; He et al., 2021), healthcare settings pose greater challenges, where labeled data are often extremely limited and test distributions differ greatly from those seen during training. In these settings, common FM adaptation methods may overfit, fail to generalize, or underperform (Imam et al., 2026; Kumar et al., 2022). As a result, there remains a need for adaptation strategies that are effective in low-data, out-of-distribution medical applications.

A promising yet underexplored alternative is in-context learning (ICL), which enables models to infer a new task directly from a small set of example inputs and outputs. While ICL has transformed natural language processing, its adoption in medical imaging has been narrow, focusing almost exclusively on segmentation. Existing visual ICL methods, including SegGPT (Wang et al., 2023), and Iris (Gao et al., 2025), demonstrate strong adaptability to unseen anatomical structures through reference image–mask pairs. However, these approaches face key limitations: (i) They are restricted to segmentation tasks, leaving classification, detection, and diagnostic reasoning largely unexplored; (ii) Many rely on computationally heavy architectures or inefficient inference (e.g., repeated reference encoding), and limiting scalability; (iii) They do not explicitly address the broader issue of clinical heterogeneity, where tasks vary not only by anatomy but also by disease definitions, scanner protocols, and institutional differences.

To overcome these limitations, we introduce Prototypical In-Context Knowledge Adaptation for Clinical Heterogeneous Usage (*PIKACHU*), a lightweight in-context learning framework designed specifically domain and classification task adaptation via in context learning on a very small number of reference image-label pairs.

*PIKACHU* enables foundation models to adapt to entirely new classification tasks using only a few reference image-label pairs (i.e., *support sets*), without fine-tuning, retraining, or relying on large annotated datasets. This framework uses a perceptual task encoding module that distills disease- and domain-specific cues from the support examples into compact,

reusable task embeddings. The embeddings are then used to condition an inference module that performs classification in a single forward pass. Since the encoder parameters are never updated, *PIKACHU* avoids catastrophic forgetting and overfitting, trains efficiently, and remains robust across varied imaging modalities and distributions. Our contributions are summarized as follows:

i. We propose a universal in-context learning framework by introducing a task encoding mechanism that captures disease- and domain-specific signals from only a few reference samples during inference, without requiring parameter updates.

ii. We perform comprehensive evaluations across heterogeneous datasets, tasks, and foundation models (including both natural and medical imaging models, as well as vision-language FMs and vision-only FMs), demonstrating improved generalization and performance to novel disease categories and distribution shifts.

iii. We conduct ablation studies to determine the impact of the support set size (i.e., number of labelled samples per class) on classification performance.

PIKACHU represents a step toward truly adaptable clinical AI, one that learns new imaging tasks instantly from examples, mirrors the flexibility of clinicians, and operates robustly in heterogeneous real-world environments.

## 2. Methodology

*PIKACHU* implements an in-context learning (ICL) pipeline designed to adapt VLFM to new classification tasks and domains using only a handful of labeled support examples. The method operates entirely in a feature-space defined by a frozen pretrained image encoder and requires learning only a single temperature parameter. This preserves the generalization capabilities learned from large-scale multimodal biomedical pre-training and makes it suitable for rapid deployment across diverse clinical workflows. Our methodology combines: (i) a frozen imaging encoder or foundation model; (ii) a prototypical in-context inference mechanism; and (iii) a lightweight learnable temperature parameter enabling task-specific calibration.

Unlike prior medical ICL work restricted to segmentation or heavy cross-attention adapters, *PIKACHU* directly performs few-shot classification using the representational geometry of a pretrained foundation model. Figure 1 summarizes our approach.

### 2.1. Problem Formulation

Given a query image $x_q$ and a small support set $\mathcal{S}$ containing $K$ labeled examples from each of $C$ classes, *PIKACHU* classifies $x_q$ using three sequential steps:

i. Extracting normalized image embeddings using a frozen foundation model vision encoder (Section 2.2);

ii. Constructing class prototypes by averaging the embeddings of support examples (i.e., for each class, we take the average of the image embeddings across all samples within that class, Section 2.3);

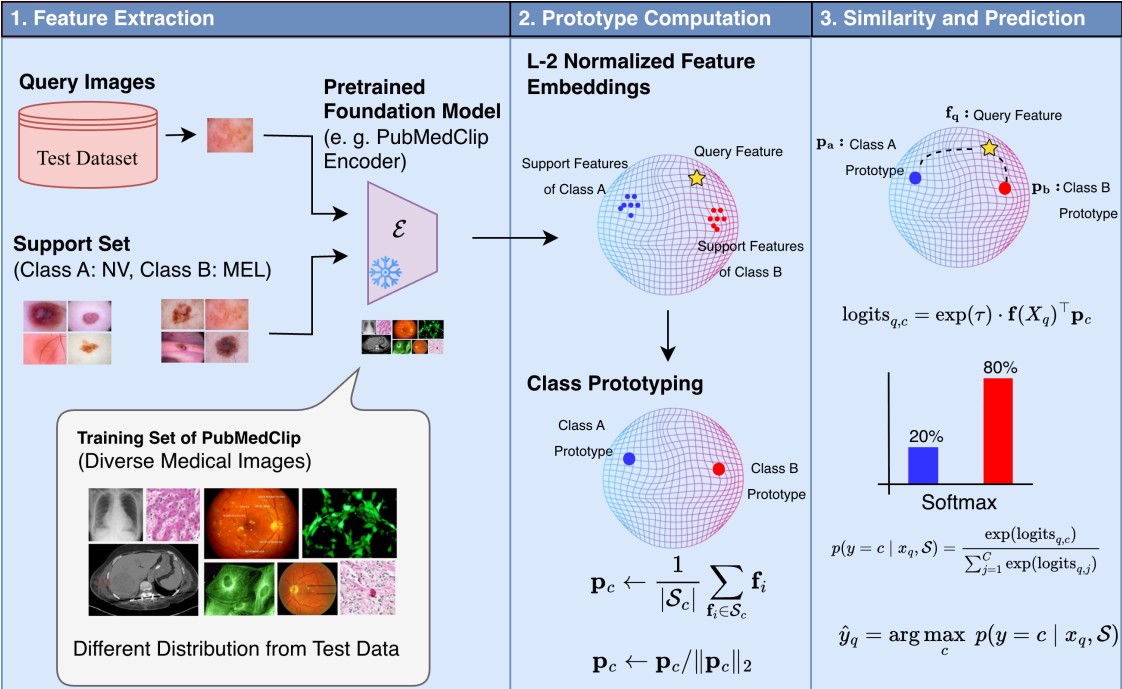

Figure 1: Overview of the *PIKACHU* framework. The method operates in three stages: (1) Feature Extraction: Query images from the test dataset and support set samples (e.g. Class A: NV, Class B: MEL) are encoded using a frozen pretrained foundation model (e.g. PubMedCLIP encoder). The encoder is pretrained on diverse medical images with distributions different from the test data. (2) Prototype Computation: L-2 normalized feature embeddings from the support set are aggregated to construct class prototypes $p_c$ through mean pooling of class-specific features. (3) Similarity and Prediction: Query features $f_q$ are compared against class prototypes using temperature-scaled cosine similarity, producing prediction probabilities via softmax computations. The final prediction $\hat{y}_q$ is determined by selecting the class with maximum posterior probability.

iii. Performing temperature-scaled cosine similarity between the query embedding and each prototype to obtain a final prediction (Section 2.4). Because only the temperature parameter is learned during training, the method maintains strong calibration and generalizes across diverse tasks without modifying the pretrained backbone.

## 2.2. Support-Query Batch Construction and Encoding

*PIKACHU* performs few-shot classification entirely through in-context prototypical reasoning. Each task consists of a *query image* and a *support image*. For a $C$-way, $K$-shot classification task, the support set is defined as

$$\mathcal{S} = \{(x_i, y_i)\}_{i=1}^{C \times K},$$

containing $K$ labeled examples from each of the $C$ classes. Support sets are constructed by grouping images by class and sampling $K$ labeled examples per class. Both support and

query images are then input to a frozen vision encoder to produce high-dimensional embeddings: $\mathbf{f}(x) = \text{Encode}(x)$, which is subsequently L2-normalized to ensure stable geometric comparisons:

$$\mathbf{f}(x) \leftarrow \frac{\mathbf{f}(x)}{\|\mathbf{f}(x)\|_2}. \tag{1}$$

These embeddings serve as the basis for prototype computation and inference.

For each task, the support set is constructed by randomly sampling K examples from each class. For a given support set size K, the sampled support set is fixed and used consistently across all corresponding evaluations, ensuring a controlled comparison while maintaining random class-balanced selection. The default value of K is 5.

## 2.3. Prototype Computation

For each class $c \in \{1, \ldots, C\}$, *PIKACHU* computes a class prototype by averaging the embeddings of the corresponding support images (note that using a different aggregation strategy in such a low data regime of 1-10 examples might lead to instability):

$$\mathbf{p}_c = \frac{1}{K} \sum_{(x_i, y_i = c)} \mathbf{f}(x_i) \tag{2}$$

To stabilize similarity computations, each prototype is further normalized:

$$\mathbf{p}_c \leftarrow \frac{\mathbf{p}_c}{\|\mathbf{p}_c\|_2} \tag{3}$$

If a class is absent in a particular episode (rare in balanced sampling), a zero vector is used by design. The resulting prototype matrix $\mathbf{P} = [\mathbf{p}_1, \ldots, \mathbf{p}_C]^\top$ forms the in-context representation against which query embeddings are compared.

## 2.4. Cosine-Similarity Inference with Temperature Scaling

Given a query embedding $\mathbf{f}(x_q)$ and the set of class prototypes $\{\mathbf{p}_c\}$, PIKACHU performs classification via temperature-scaled cosine similarity:

$$\text{logits}_{q,c} = \exp(\tau) \cdot \mathbf{f}(x_q)^\top \mathbf{p}_c \tag{4}$$

where $\tau$ is a learnable temperature parameter (optimized during training) that sharpens or smooths the similarity distribution. Since both embeddings and prototypes lie on the unit hypersphere, the dot product is equivalent to cosine similarity.

The final class probabilities are obtained by applying a softmax:

$$p(y_q = c \mid x_q, \mathcal{S}) = \frac{\exp(\exp(\tau) \cdot \mathbf{f}(x_q)^\top \mathbf{p}_c)}{\sum_{c'=1}^{C} \exp(\exp(\tau) \cdot \mathbf{f}(x_q)^\top \mathbf{p}_{c'})} \tag{5}$$

Crucially, the temperature $\tau$ is the *only* trainable parameter in the entire framework. This ensures that PIKACHU maintains the in-context property of few-shot learning while requiring minimal computation and avoiding any task-specific fine-tuning of the foundation model. Algorithm 1 (Appendix A) shows the detailed steps for our implementation.

## 3. Experiments and Results

### 3.1. Dataset and Implementation Details

#### 3.1.1. Datasets

We evaluate our approach on three diverse medical imaging datasets spanning different anatomical regions and diagnostic tasks to assess the generalizability of ICL across various clinical scenarios.

*Skin Dataset (ISIC)* (Tschandl et al., 2018; Codella et al., 2018; Combalia et al., 2019): This dataset comprises dermatological images for skin lesion classification, including multiple categories of benign and malignant conditions such as melanoma, basal cell carcinoma, actinic keratosis, and benign nevi. The dataset presents challenges due to significant intra-class variability in lesion appearance, color, texture, and morphology across different skin types and imaging conditions. We perform binary classification between melanoma (MEL) and melanocytic nevi (NV) with 4522 samples in each test class.

*OCT (Optical Coherence Tomography)* (Kermany et al., 2018): The OCT dataset contains cross-sectional retinal scans used for diagnosing various retinal pathologies. Images include normal retinal tissue as well as pathological conditions such as diabetic macular edema (DME), choroidal neovascularization (CNV), and drusen. OCT imaging provides high-resolution visualization of retinal layers, making it valuable for early detection and monitoring of retinal diseases. We perform binary classification between CNV and Normal scans with 1000 samples per class.

*DR (Diabetic Retinopathy)* (Eyepacs, 2025): This dataset consists of fundus photographs for diabetic retinopathy screening and grading. Images are categorized based on DR severity levels ranging from no DR to Proliferative Diabetic Retinopathy (PDR), with intermediate stages including mild, moderate, and severe non-proliferative DR. The task requires identifying subtle vascular abnormalities, microaneurysms, hemorrhages, and neovascularization. We perform binary classification between no DR and Proliferative Diabetic Retinopathy (PDR) with 1466 samples from each test class.

For all three datasets, we formulate the evaluation as binary classification tasks to facilitate consistent comparison across different medical imaging modalities. Importantly, the test sets are carefully balanced to ensure equal representation of positive and negative classes, eliminating class imbalance as a confounding factor in performance evaluation and providing a fair assessment of model discrimination capabilities.

#### 3.1.2. Implementation Details

To contextualize our implementation, we briefly outline the foundation models used in our experiments. Evaluating PIKACHU across these backbones allows us to assess the generality of our in-context learning strategy under differing pretraining objectives, data sources, and representational characteristics:

i. *PubMedCLIP* (Eslami et al., 2023) is a vision-language model pretrained on large-scale biomedical image–text pairs, providing domain-specific representations well suited for clinical imagery; its medical grounding makes it a strong baseline for specialized tasks.

ii. *SigLIP (Sigmoid Loss for Language-Image Pre-Training)* (Zhai et al., 2023) is a contrastive vision–language model trained on massive web-scale datasets using a sigmoid loss formulation, enabling robust and semantically aligned visual features that generalize effectively beyond its training distribution.

iii. *DINOv2* (Oquab et al., 2023) is a self-supervised vision model learned through knowledge distillation without labels, trained on a huge dataset of web-curated images and producing highly transferable visual embeddings that excel across diverse downstream tasks.

iv. *Vision Transformer (ViT)* (Wu et al., 2020) represents a purely vision-based architecture trained on generic large-scale image datasets, offering a neutral baseline with no modality-specific inductive biases.

All experiments are implemented in PyTorch, with pretrained foundation model backbones loaded through their respective public model hubs. Model weights are downloaded once and cached locally for reproducibility and efficient re-use. Each backbone is kept fully frozen throughout training and inference to preserve its pretrained representations and ensure a consistent evaluation of in-context adaptation. All input images are converted to PIL format and preprocessed using the normalization pipeline associated with each model before feature extraction.

The only trainable parameter is the logarithm of the temperature $\log T$, optimized with Adam using a learning rate of $1 \times 10^{-4}$. Training is performed for 5-10 epochs depending on the dataset, with each epoch consisting of thousands of randomly sampled few-shot episodes. Evaluation uses the identical episodic structure but with no gradient updates. Metrics such as accuracy, AUROC, $F_1$-score, and confusion matrices are computed to assess performance.

All experiments are conducted on a single NVIDIA H100 GPU, although CPU inference is feasible due to the minimal number of trainable parameters.

### 3.2. Experiments

Our unified experimental design isolates the contribution of in-context learning itself, allowing us to compare improvement margins across architectures with differing representational strengths. For each of the four foundation models above, the baseline setting corresponds to the standard zero-shot or fixed-feature configuration, where the encoder remains frozen and predictions are produced without any access to support examples. This involves computing cosine similarity between query embeddings and predefined classifier weights (see Appendix B). In the baseline settings, no task-specific adaptation or additional fine-tuning is performed.

### 3.3. Results & Ablations

**Baseline v/s ICL strategy**  Table 1 demonstrates that ICL consistently and substantially outperforms baseline approaches across all vision models and medical imaging datasets. The improvement is particularly pronounced in the OCT dataset, where models like SigLip achieve a remarkable jump from 0.50 to 0.83 accuracy. Similarly, PubMedCLIP shows significant gains in OCT performance (0.50 to 0.72 accuracy) and DR classification (0.40 to

0.79 accuracy), highlighting ICL's effectiveness in medical image analysis tasks. Interestingly, domain-specific models like PubMedCLIP and general vision models like DinoV2 and ViT all benefit substantially from ICL. The skin data classification shows more modest but consistent improvements (0.19-0.24 accuracy gain), while OCT and DR datasets exhibit the most dramatic performance enhancements. These findings suggest that ICL is particularly effective for complex medical imaging tasks where subtle pathological features need to be distinguished, demonstrating its potential as a powerful strategy for improving diagnostic accuracy without requiring extensive model retraining or fine-tuning.

| Model | Strategy | Dataset | | |
| | | ISIC | OCT | DR |
| --- | --- | --- | --- | --- |
| SigLIP | Baseline | 0.49 | 0.50 | 0.50 |
| | ICL | 0.73 | 0.83 | 0.77 |
| PubMedCLIP | Baseline | 0.50 | 0.50 | 0.40 |
| | ICL | 0.69 | 0.72 | 0.79 |
| DinoV2 | Baseline | 0.50 | 0.51 | 0.76 |
| | ICL | 0.74 | 0.82 | 0.80 |
| ViT | Baseline | 0.50 | 0.52 | 0.39 |
| | ICL | 0.69 | 0.83 | 0.81 |

Table 1: Performance comparison of different vision models (SigLip, PubMedCLIP, DinoV2, and ViT) using baseline(zero-shot) and ICL (In-Context Learning) strategies across three medical imaging datasets: ISIC, OCT (Optical Coherence Tomography), and DR (Diabetic Retinopathy).

Figure 2 shows the confusion matrices illustrating the binary classification performance for all four FMs on each of the medical imaging datasets, with and without the proposed ICL method. Note the improvement in performance in all cases, as depicted visually by the confusion matrices moving from off diagonal (baseline) to diagonal (*PIKACHU*).

**Effect of Support Queries** To investigate the impact of support set size on ICL performance, we conducted experiments varying the number of support samples per class across 1, 5, and 10 examples. The results reveal a clear positive correlation between the number of support queries and model performance across all three medical imaging datasets. With a single support sample per class, models achieved modest improvements over baseline performance, demonstrating that even minimal context can enhance classification accuracy. Increasing the support set to 5 samples per class yielded substantial performance gains, particularly evident in the OCT and DR datasets, where the additional examples provided richer context for distinguishing between subtle pathological variations. At 10 support samples per class, we observed further improvements, though with diminishing returns compared to the jump from 1 to 5 samples, suggesting a saturation effect where additional examples provide incrementally less new information.

### 3.4. Limitations

While *PIKACHU* provides a simple and effective mechanism for few-shot adaptation in medical image classification, several limitations warrant consideration. First, the frame-

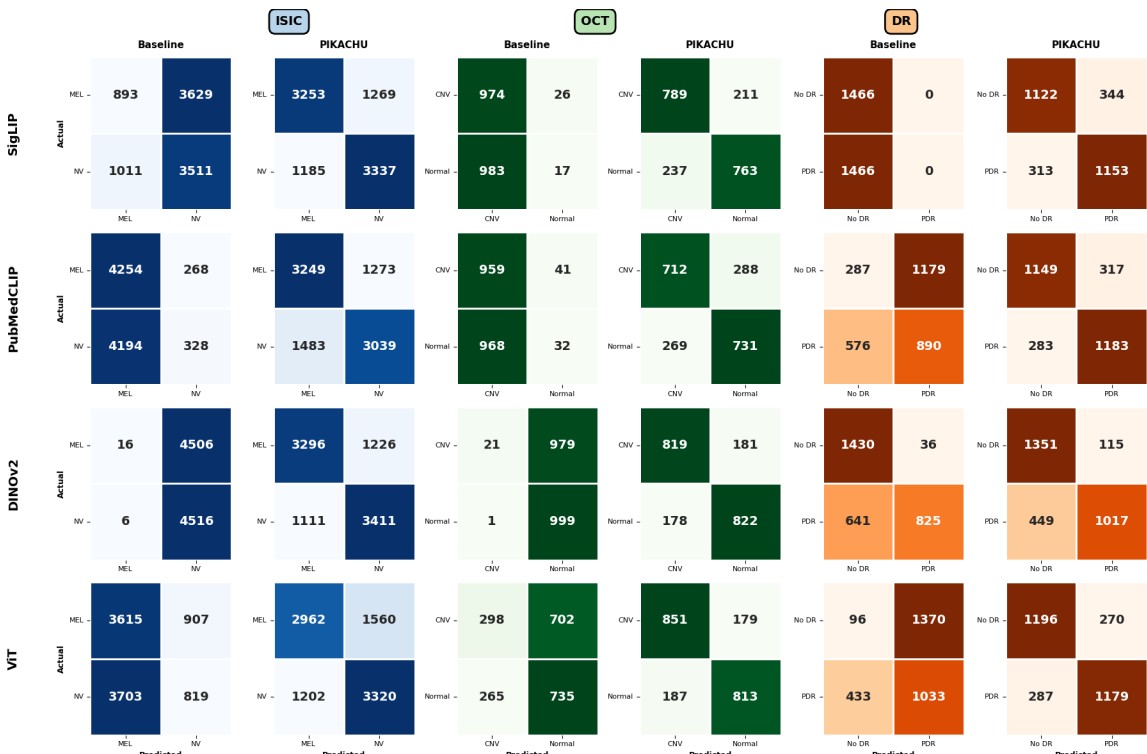

Figure 2: Confusion matrices comparing the binary classification performance of the four FMs on three medical imaging tasks on different datasets (ISIC, OCT, DR). For each dataset pair, the first confusion matrix (left) represents the baseline performance, while the second (right) shows results using the ICL method (*PIKACHU*) with K=5. Notice that for each confusion matrix pair, we observe a clear redistribution of predictions from off-diagonal elements in the baseline to diagonal elements with the proposed ICL method, indicating a substantial reduction in misclassification rates and improved class-wise discrimination.

work relies on the representational quality of the underlying frozen backbone. If the pre-trained features fail to capture clinically meaningful disease cues, prototype construction may be insufficient to recover task-specific distinctions. Second, the method assumes that a small number of support examples are representative of each class, which may not hold in settings with extreme intra-class variability or subtle disease patterns. Additionally, our current formulation treats each support example independently and does not account for label noise, class imbalance, or image acquisition artifacts commonly encountered in clinical datasets. Prototype averaging may also oversimplify complex class manifolds, particularly for multi-modal or heterogeneous disease categories, where more expressive task embeddings could yield further gains. Finally, our experiments focus on episodic evaluation under controlled few-shot settings. Additional work is needed to assess robustness under real-world deployment scenarios, such as varying support set quality, shifting patient populations, or incomplete class coverage. Despite its robustness, these limitations outline opportunities for improving the adaptability and reliability of in-context learning in medical imaging.

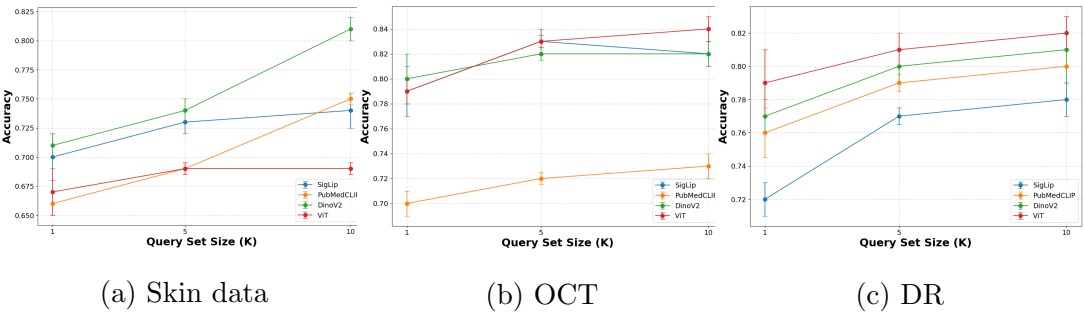

| (a) Skin data | (b) OCT | (c) DR |

Figure 3: Impact of support set size (K) on ICL performance. Accuracy increases with more support samples per class (K = 1, 5, 10) across all datasets and models, with steeper improvements in OCT and DR compared to Skin data. Performance gains are most pronounced between K=1 and K=5.

.

## 4. Conclusion

In this work, we presented *PIKACHU*, a lightweight and universal in-context learning framework for medical image classification. By operating entirely in the feature space of frozen foundation models and leveraging a simple yet effective prototypical reasoning mechanism, *PIKACHU* enables rapid adaptation to new clinical tasks using only a few labeled examples. Our method requires no fine-tuning, no retraining, and only a single learned temperature parameter, making it computationally efficient and easy to deploy across heterogeneous clinical environments. Experiments across multiple backbones, including PubMedCLIP, SigLIP, DINOv2, and ViT, demonstrate that incorporating a small support set consistently improves performance over standard zero-shot or frozen-feature baselines by just using 5 samples from each class. These findings highlight the potential of in-context learning as a practical and scalable pathway for adapting foundation models to the varied and evolving demands of real-world medical imaging workflows.

While *PIKACHU* offers a simple and effective solution for few-shot medical image classification, several promising directions remain for future research. First, extending in-context adaptation to more complex clinical tasks such as multi-label disease prediction, longitudinal progression modeling, or structured report generation may further expand the applicability of this framework. Second, exploring richer task representations, including textual or multimodal prompts, may enhance prototype quality and improve robustness under severe distribution shifts. Third, integrating uncertainty estimation or confidence calibration into the ICL pipeline could make the method more suitable for safety-critical scenarios. Finally, evaluating *PIKACHU* across larger-scale hospital datasets and under real-world operational constraints would provide deeper insight into its practical utility. Together, these directions offer a path toward more adaptive, reliable, and generalizable AI systems for clinical imaging.

## Acknowledgements

The authors are grateful for funding provided by the Natural Sciences and Engineering Research Council of Canada, the Canadian Institute for Advanced Research (CIFAR) Artificial Intelligence Chairs program, Mila - Quebec AI Institute, Google Research, Calcul Quebec, Fonds de recherche du Québec (FRQNT), the Digital Research Alliance of Canada, and the Vadasz Scholar McGill Engineering Doctoral Award.

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

## Appendix A. *PIKACHU* Algorithm

In this section, we present a detailed, stepwise description of the *PIKACHU* in-context learning framework. The algorithm formalizes the complete workflow of our method, beginning with support-query preparation, followed by prototype construction in the frozen foundation-model feature space, and concluding with temperature-scaled similarity inference.

---

**Algorithm 1 PIKACHU: Perceptual In-Context Knowledge Adaptation for Medical Image Classification**

---

**Input:** Frozen Foundation Model such as PubMedCLIP encoder $\mathcal{E}$;
Support set $\mathcal{S} = \{(x_i, y_i)\}$ with $C$ classes and $K$ shots per class;
Query image $x_q$;
Learnable temperature parameter $\log T$.
**Output:** Predicted label $\hat{y}_q$ for the query image.

**Step 1: Support-Query Preprocessing and Feature Extraction**
  Convert all support and query images to PIL format and preprocess/transforms.
**foreach** *support image $x_i$* **do**
  $\mathbf{f}_i \leftarrow \mathcal{E}(x_i)$                           // Frozen encoder
  $\mathbf{f}_i \leftarrow \mathbf{f}_i / \|\mathbf{f}_i\|_2$                           // L2-normalize
**end**
$\mathbf{f}_q \leftarrow \mathcal{E}(x_q); \mathbf{f}_q \leftarrow \mathbf{f}_q / \|\mathbf{f}_q\|_2$                           // Query embedding

**Step 2: Prototype Computation**
  Initialize prototype vectors $\mathbf{p}_1, \ldots, \mathbf{p}_C$.
**for** $c = 1$ **to** $C$ **do**
  $\mathcal{S}_c \leftarrow \{\mathbf{f}_i \mid y_i = c\}$                           // Support features of class $c$
  **if** $\mathcal{S}_c$ *not empty* **then**
    $\mathbf{p}_c \leftarrow \frac{1}{|\mathcal{S}_c|} \sum_{\mathbf{f}_i \in \mathcal{S}_c} \mathbf{f}_i$                           // Mean prototype
    $\mathbf{p}_c \leftarrow \mathbf{p}_c / \|\mathbf{p}_c\|_2$                           // Normalize
  **else**
    $\mathbf{p}_c \leftarrow \mathbf{0}$                           // Fallback for missing class
  **end**
**end**

**Step 3: Temperature-Scaled Cosine Similarity Inference**

Compute logits for each class:

$$\text{logits}_c = \exp(\log T) \cdot (\mathbf{f}_q^\top \mathbf{p}_c)$$

Apply softmax:

$$p(y = c \mid x_q, \mathcal{S}) = \frac{\exp(\text{logits}_c)}{\sum_{j=1}^{C} \exp(\text{logits}_j)}$$

Predict class:

$$\hat{y}_q = \arg\max_c \ p(y = c \mid x_q, \mathcal{S})$$

**Training Procedure (Episodic Few-Shot)**
  During training, only *one parameter* is optimized: $\log T$.
**foreach** *episode* **do**
  Construct support set $\mathcal{S}$ and query batch. Compute prototypes and logits as above.
    Update $\log T$ using cross-entropy loss and Adam optimizer.
**end**
**return** $\hat{y}_q$

---

# Appendix B. Additional Implementation Details

## B.1. Training using DINOv2, ViT

**DINOv2**  In this DINOv2-based setup the backbone is kept fully frozen and classification is performed via a prototypical, similarity-based head. Images are first encoded using a frozen DINOv2-Large model, and the resulting CLS-token features are L2-normalized. Class prototypes are then computed on-the-fly as the mean of support features for each class, followed by normalization. During training, gradients flow exclusively to the temperature parameter, while the backbone and prototype construction remain fixed. This design enables stable and efficient adaptation by calibrating decision boundaries without learning a conventional linear layer or modifying the pretrained DINOv2 representations.

**ViT**  Training in this setup follows an episodic in-context learning (ICL) regime that mirrors few-shot inference rather than conventional supervised fine-tuning. For each training batch, episodes are constructed consisting of a support set and corresponding query samples. Query and support images are first passed through a frozen ViT-Base backbone to extract L2-normalized CLS-token features. Class prototypes are then computed on-the-fly as the mean of the support features for each class within the episode. Classification logits for query samples are obtained via cosine similarity to these prototypes, scaled by a learnable temperature. During backpropagation, only the temperature parameter is updated using cross-entropy loss on the query labels, while the backbone and prototype computation remain fixed. This episodic training calibrates the similarity scale across tasks without altering the representation space, ensuring stable optimization and consistency between training and inference in the few-shot setting

# Appendix C. Ablations

## C.1. Comparisons to other few-shot classification methods

While PIKACHU operates through in-context prototypical reasoning with a single learnable temperature parameter, we contextualize our approach by comparing against three representative methods: Tip-Adapter (Zhang et al., 2022), Proto-Adapter (Kato et al., 2024), and LoRA (Hu et al., 2022).

**Tip-Adapter:**  Tip-Adapter (Zhang et al., 2022) augments frozen CLIP-style models with a cache of support embeddings and performs prediction via weighted retrieval, optionally combining cache-based scores with zero-shot logits. The training-free variant requires no optimization, while the trainable variant learns cache keys with $N \times K \times D$ parameters. However, inference requires computing similarities against all cached support samples, leading to $\mathcal{O}(N \times K)$ complexity and linear memory growth with the support set size. In contrast, PIKACHU compresses all support information into $C$ class prototypes, enabling $\mathcal{O}(C)$ inference and a substantially lower memory footprint.

**Proto-Adapter:**  Proto-Adapter (Kato et al., 2024) introduces lightweight bottleneck modules that refine features through $\text{Adapter}(\mathbf{f}) = \mathbf{W}_{\text{up}} \cdot \text{GELU}(\mathbf{W}_{\text{down}} \cdot \mathbf{f})$ with dimensionality reduction ($D \to d \to D$, typically $d = 64$). This yields $2 \times D \times d + 1$ trainable parameters, enabling task-specific feature space transformations. While this expressiveness

may benefit tasks where pretrained features are misaligned, it introduces more parameters than PIKACHU and risks overfitting in extreme low-data regimes ($K \leq 5$). PIKACHU's design philosophy instead trusts the discriminative power of modern medical foundation models (e.g., PubMedCLIP (Eslami et al., 2023)), requiring no feature manipulation.

**LoRA:** LoRA (Hu et al., 2022), originally developed for large language models, adapts frozen weights through low-rank decompositions: $\mathbf{W}' = \mathbf{W} + \frac{\alpha}{r}\mathbf{BA}$ where $\mathbf{A} \in \mathbb{R}^{r \times D}$, $\mathbf{B} \in \mathbb{R}^{D \times r}$, and $r \ll D$. With rank $r \in \{8, 16\}$, LoRA introduces $2 \times r \times D + 1 \approx 12$–25K parameters. While LoRA demonstrates strong empirical success in NLP and offers a compelling balance between simplicity and expressiveness, it still requires two orders of magnitude more parameters than PIKACHU and necessitates careful rank selection. Additionally, LoRA's injection of low-rank updates into weight matrices may inadvertently disrupt pretrained medical knowledge when support sets are limited.

| Few-Shot Method | Datasets | | | Trainable Params |
|---|---|---|---|---|
| | **ISIC** | **OCT** | **DR** | |
| Tip-Adapter | 0.71 | 0.78 | 0.74 | 1 |
| Proto-Adapter | **0.73** | 0.82 | 0.76 | 99,137 |
| Lora (rank=8) | 0.72 | **0.83** | 0.76 | 12,289 |
| Lora (rank=16) | 0.73 | 0.82 | **0.77** | 24,577 |
| PIKACHU | **0.73** | **0.83** | **0.77** | 1 |

Table 2: Few-shot comparison showing that PIKACHU attains competitive performance relative to Tip-Adapter, Proto-Adapter, and LoRA, despite introducing only a single trainable parameter. Note SigLIP is the backbone foundation model here.

Table 2 summarizes the architectural and computational trade-offs. PIKACHU occupies an extreme point in the PEFT design space, sacrificing feature-space adaptation for parameter efficiency. We show that PIKACHU achieves performance on par with, and in some cases matching the best results of, stronger adaptation methods while being more parameter-efficient. Across ISIC, OCT, and DR, PIKACHU matches the top accuracy achieved by Proto-Adapter and LoRA variants, despite introducing only a single trainable temperature parameter. In contrast, competing methods require orders of magnitude more parameters (e.g., tens of thousands for LoRA and nearly 100k for Proto-Adapter). This highlights that much of the few-shot performance gain can be attributed to effective prototype construction and similarity calibration rather than extensive parameter adaptation, demonstrating that PIKACHU provides an excellent accuracy–efficiency trade-off.

### C.2. In-Context Learning Aggregation Strategy: K-Nearest Neighbors with Weighted Aggregation

While the prototypical network approach (described in Section 2.3) has demonstrated strong performance for few-shot medical image classification, offering an elegant and computationally efficient solution through class-level prototype representations. However, for completeness and to provide a comprehensive evaluation of in-context learning paradigms, we conduct an ablation study that explores alternative aggregation strategy, *K-nearest neighbors with weighted aggregation*. This alternative preserve fine-grained support set structure rather

than collapsing samples into class-level prototypes, which may be beneficial in edge cases involving complex multimodal distributions or overlapping decision boundaries.

Unlike prototypical networks, which condense all support samples of a given class into a single representative vector, the K-nearest neighbors (KNN) approach with weighted aggregation preserves the contribution of individual support examples. This enables the model to adapt its predictions based on local similarity structure rather than relying solely on global class prototypes.

**Feature Extraction and Similarity Computation**   Given a query image $x_q$ and support set $\mathcal{S} = \{(x_i, y_i)\}_{i=1}^{N \times K}$, we first extract and normalize features using the frozen encoder $E(\cdot)$ as in the prototypical network:

$$\mathbf{f}_i = \frac{E(x_i)}{\|E(x_i)\|_2}, \quad \mathbf{f}_q = \frac{E(x_q)}{\|E(x_q)\|_2} \tag{6}$$

For each query feature $\mathbf{f}_q$, we compute cosine similarities to all support features:

$$s_i = \mathbf{f}_q^\top \mathbf{f}_i, \quad \forall (x_i, y_i) \in \mathcal{S} \tag{7}$$

Since both query and support features are L2-normalized, this dot product directly corresponds to cosine similarity.

**K-Nearest Neighbor Selection.**   We identify the $K$ support samples with the highest similarity scores:

$$\mathcal{N}_K(x_q) = \{(x_{i_1}, y_{i_1}), \dots, (x_{i_K}, y_{i_K})\} \tag{8}$$

where $s_{i_1} \geq s_{i_2} \geq \cdots \geq s_{i_K}$ denote the top-$K$ similarity scores among all support samples. This selection process focuses attention on the most relevant support examples for each query.

**Distance-Weighted Aggregation.**   Rather than treating all $K$ neighbors equally, we employ distance-weighted voting where each neighbor's contribution is proportional to its similarity to the query. The weights are computed via a temperature-scaled softmax:

$$w_j = \frac{\exp(s_{i_j}/\tau)}{\sum_{k=1}^K \exp(s_{i_k}/\tau)} \tag{9}$$

where $\tau$ is a learnable temperature parameter (optimized as $\log \tau$ during training).

The final class probabilities are obtained by aggregating weighted votes across the $K$ nearest neighbors:

$$P(y = c \mid x_q, \mathcal{S}) = \sum_{j=1}^K w_j \cdot \mathbb{1}[y_{i_j} = c] \tag{10}$$

where $\mathbb{1}[\cdot]$ denotes the indicator function.

Table 3 compares different in-context aggregation strategies using a fixed SigLIP backbone. The baseline zero-shot performance is near chance across all datasets, while KNN-weighted in-context learning provides a clear improvement by leveraging support examples. PIKACHU further yields a substantial performance gain on ISIC, OCT, and DR, consistently outperforming both the baseline and KNN-based ICL. This demonstrates that prototype-based aggregation with similarity calibration is significantly more effective than instance-level retrieval for in-context adaptation in medical imaging tasks.

| Model | Strategy | Datasets | | |
| --- | --- | --- | --- | --- |
| | | **ISIC** | **OCT** | **DR** |
| | Baseline | 0.49 | 0.50 | 0.50 |
| SigLIP | ICL (KNN weighted) | 0.61 | 0.64 | 0.70 |
| | PIKACHU | **0.73** | **0.83** | **0.77** |

Table 3: Performance of different aggregation strategies, prototype- and KNN-based, for in-context learning (ICL) on ISIC, OCT, and DR datasets.

