# OpenReview forum: "PIKACHU: Prototypical In-context Knowledge Adaptation for Clinical Heterogeneous Usage"
_MIDL.io/2026/Conference — MIDL 2026 Poster_

### Official Review · Reviewer_5chs · 2026-01-07

**Confidence:** 4
**Preliminary Rating:** 4
**Final Rating:** 4

**Summary:**

PIKACHU focuses on in-context learning framework for classification of medical images. This work uses a small support set in inference to construct class specific prototypes for few shot learning with VLFM’s. Catastrophic forgetting is overcome with the usage of a single temperature parameter used for similarity measurement.

**Strengths:**

- The authors present a very well structured and well written manuscript. The framework is clearly explained with details provided for reproducibility.
- Simple but effective method to improve performance of VLFM’s in unseen inference tasks.
- The method is compared against different backbone VLFM’s across diverse images.

**Weaknesses:**

- As identified by the authors themselves, evaluation has been oversimplified by considering only binary classification on balanced datasets. This is not a faithful reproduction of a real world inference scenario. While comparison across datasets with multiclass classification or class imbalance would be harder without domain knowledge, this would be necessary to understand how this simple framework performs under different conditions.
- Different aggregation strategies could have been presented as an ablation study.

**Detailed Comments:**

The authors haven’t provided details on how the support set was constructed. Was it a random sampling? Could that explain the lack of performance improvement going from 5 to 10 examples in the support set? These questions also lose some value for a simple balanced, binary classification inference scenario but would still be useful.

**Justification Of Final Rating:**

I appreciate the authors’ efforts in addressing my comments from the preliminary review and thank them for clarifying the construction of the support set. While starting with a balanced binary task is a reasonable choice to isolate the impact of the proposed prototype selection, I believe the evaluation remains limited and could be strengthened to better assess the application of the work in real-world setting. Accordingly, I will maintain my score of 4.

**Justification Of The Preliminary Rating:**

The authors present a novel, simple and effective method for few shot learning with VLFM’s. However, the evaluation setup limits the overall impact of the work. A more challenging and comprehensive evaluation would strengthen assessment of the method. As acknowledged by the authors, the experiments focus exclusively on binary classification with balanced datasets, which does not fully reflect real-world medical imaging inference scenarios. While extending the evaluation to multiclass or class-imbalanced settings would be more challenging and potentially require additional domain knowledge, it would be important to assess the robustness of the framework. An ablation study for different aggregation strategies and sufficient detail on construction of the support set can help with a more complete evaluation.

**Questions To Address In The Rebuttal:**

Already stated above.

---

> ### Author Response · Authors · 2026-01-25
>
> We would like to sincerely thank the reviewer for their positive feedback on our paper. We are pleased to hear that they find our work well written and easy to follow. The reviewer requested several additional experiments and additional details on constructing the support set, which are addressed below:
> 1. **Extending to Class-Imbalanced and Multiclass Settings**
> In this initial work, we present the first controlled and systematic study of in-context prototype construction with SOTA vision and vision-language foundation models. We therefore focus on balanced binary tasks to isolate the effects of the proposed similarity formulation and temperature scaling, without introducing additional confounders such as skewed class priors or dataset-specific heuristics. Importantly, the experimental study is already extensive, spanning multiple foundation models, three distinct medical imaging modalities, and a wide range of support-set sizes, resulting in a large number of experimental configurations (72 experiments in the submitted paper and 15 additional experiments during the rebuttal). Beginning with the task of binary classification permits a clear understanding of the core behavior of PIKACHU before expanding along additional axes of complexity (e.g. as introduced by multi-class or class-imbalanced classification).
>
>     We do, however, acknowledge that the real world scenario is more complex. Therefore, to further demonstrate generality, we have added supplementary experiments in the appendix C comparing against other Few-shot methods (Tip-adapter, Proto-adapter and Lora) and other aggregation strategy (KNN). The results are consistent with the trends reported in the main paper.  We would like to emphasise that PIKACHU does not assume class balance or binary labels by design; extending to the more complex settings (e.g. multi-class) only requires modifying support-set composition and evaluation metrics, without requiring changes to the underlying framework.
>
> 2. **Details on Constructing the Support Set**
> We have added details on page 5 regarding how the support set was constructed:
>     ```For each task, the support set is constructed by randomly sampling K examples from each class. For a given support set size K, the sampled support set is fixed and used consistently across all corresponding evaluations, ensuring a controlled comparison while maintaining random class-balanced selection. The default value of K is 5.```
>
> 3. **Effect of Support Set Size on Performance**
> The reviewer raises an insightful question regarding the limited performance improvement observed when increasing the support set size from 5 to 10 examples. Indeed, the results indicate that a small number of representative examples is often sufficient to estimate stable class prototypes. In such cases, adding more samples may yield diminishing returns rather than indicating an issue with random sampling. In more challenging scenarios such as class imbalance, higher intra-class variability (multi-modal distributions), or multiclass tasks, the impact of increasing the support set size and sampling strategy would likely be more pronounced.

---

### Official Review · Reviewer_WGZg · 2026-01-10

**Confidence:** 3
**Preliminary Rating:** 2
**Final Rating:** 2

**Summary:**

The paper proposes an in-context learning method for foundation models for medical image classification. Specifically, it does few-shot classification by constructing class prototypes from embeddings obtained from frozen foundation/pretrained models as class centroids of some labeled samples per class and then assesses the cosine similarity of new samples to these class centroids / prototypes. Class-wise similarity is further transformed using a softmax to obtain class probabilities. The approach is compared on three different datasets for a balanced, binary classification setting with four different pretrained/foundation models. Baselines are zero-shot (prompt-based?) classification for VL FMs and linear probing for Vision Models. An ablation study investigates the impact of labeled samples per class. Strong performance improvements compared to the baselines are reported.

**Strengths:**

- The storyline set out in the paper is easy to follow and clearly laid out, addressing a potentially interesting task of adaptation for specific situations / patients etc. in low(er)-resource settings.
- The proposed approach is evaluated on three different dataset and datatypes (dermatological images, OCT, fundus photography) from publicly available sources.
- Different foundation models/pretrained models (including vision-only and vision-language models) are included in the comparison.

**Weaknesses:**

- The core contribution, in-context prototypical reasoning, boils down from my understanding to centroid classification. Generally, centroid classification and related few-shot methods (k-neighest neighbor, linear probing) in an embedding space are common ways to perform classification with a (small) set of labeled samples without additional training [see references below]. With centroid classification being an old technique as such, I fail to identify a core methodological novelty for this as in-context learning method.
- It is claimed that in-context learning is restricted to segmentation tasks. I see an issue with this statement if centroid classification is seen as in-context learning since there are many papers doing linear probing, k-nn-based classification, prompt tuning, etc. (see references below).
- The purpose of learning the temperature parameter is not clear to me, as it will not change any classification. Calibration is mentioned but this part is not really explored and I would have liked to see a corresponding ablation study.
- From my perspective, the experiments are not fully comprehensive. Important similar approaches (k-nn, CoOp) are not included and the performance of the results compared to proposed method gives rise for doubt. Looking at the accuracy and AUC, the baseline methods all have a performance below chance. Furthermore, few-shot learning performance is naturally prone to be dependent on the samples selected. It was not clear to me whether repeated experiments were conducted.
- Lastly, the paper is motivated by challenges around clinical heterogeneity (disease definitions, scanner protocols, and institutional differences); however, these aspects are not addressed in the evaluation but the results are still on "clean" datasets without these issues.

References:
General:
- Thomas Mensink, Jakob Verbeek, Florent Perronnin, Gabriela Csurka. Distance-based image classification: generalizing to new classes at near-zero cost. IEEE TPAMI, 35(11):2624–2637, 2013. DOI: 10.1109/TPAMI.2013.83.
- Jake Snell, Kevin Swersky, Richard Zemel. Prototypical Networks for Few-shot Learning. NeurIPS 2017.
Medical focus:
- Ali, S., Bhattarai, B., Kim, TK., Rittscher, J. (2020). Additive Angular Margin for Few Shot Learning to Classify Clinical Endoscopy Images. In: Liu, M., Yan, P., Lian, C., Cao, X. (eds) Machine Learning in Medical Imaging. MLMI 2020. Lecture Notes in Computer Science(), vol 12436. Springer, Cham. https://doi.org/10.1007/978-3-030-59861-7_50
- Doerrich, Sebastian; Archut, Tobias; Di Salvo, Francesco; Ledig, Christian (2024): Integrating kNN with Foundation Models for Adaptable and Privacy-Aware Image Classification, in: 2024 IEEE International Symposium on Biomedical Imaging (ISBI), IEEE, pp. 1–5, doi: 10.1109/isbi56570.2024.10635560.
- Taha Koleilat, Hojat Asgariandehkordi, Hassan Rivaz, Yiming Xiao; Proceedings of the IEEE/CVF Conference on Computer Vision and Pattern Recognition (CVPR), 2025, pp. 14766-14776
- J Qiu, N Jain, J Ammeling, M Aubreville, K Breininger; Effortless Vision-Language Model Specialization in Histopathology without Annotation, MICCAI 2025 Workshop COMPAYL 2025

**Detailed Comments:**

(not relevant for acceptance / rejection: I like the acronym)
Main comments:
- pg. 2: "After pre-training, such downstream prediction tasks (e.g., classification) can be performed through zero-shot inference in vision-language FMs, or by training simple classifiers on top of frozen or fine-tuned vision FMs." - There is quite a large number of ways to use both VL FMs and vision-only FMs, including working with frozen embeddings or fine-tuning, but since there also other methods like low-rank adaptations (e.g., LoRA [r1]), prompt tuning (e.g., Context Optimization - CoOp [r2]) this overview seems rather incomplete. This concern also translates to the methods used as comparison methods in this paper.
- pg. 3: "Because only the temperature parameter is learned during training, the method maintains strong calibration and
generalizes across diverse tasks without modifying the pretrained backbone." - What does it mean that the method "maintains strong calibration"? To my understanding, this is not evaluated.
- pg. 6: (see weaknesses) A binary classification on fairly narrow datasets seems like a highly simplified task given the broad application targeted in the introduction. It is not clear why these experiments support the broad target application laid out in the introduction.
- pg. 7: "Depending on the model, this involves either computing cosine similarity between query embeddings and predefined
classifier weights (PubMedCLIP, SigLIP)" - two concerns here: 1) I am not sure what is meant here by "predefined classifier weight"? Prompts? If yes, what were the used prompts? I further recommend to include prompt tuning and also linear probing (and other few-shot techniques as mentioned above) for VL FMs for fair comparison.
- pg. 7: "training a simple linear classifier on top of frozen features (DINOv2, ViT)" - how was the linear classifier trained?
- pg. 8 Table 1 / pg. 9 / Fig. 2: Under which setting was the evaluation on pg. 8 performed? Are the results for SigLIP correctly listed?
- prior text and pg. 10 Fig. 3: As discussed above, few-shot learning is prone to sampling dependency for the labeled samples. Especially in this evaluation I would have expected to see a performance range. Were multiple experiments with different samples performed?

Minor comments:
- pg. 2: "We conduct ablation studies to determine the impact of the support set size (i.e., number of labeled samples per class) classification performance." - typo
- pg. 5: "If a class is absent in a particular episode (rare in balanced sampling), a zero vector is used by design." - I assume that this is not applicable for any of the experiments reported here?
- pg. 5 / Fig. 1: the temperature parameter is somewhat inconsistently defined as $\tau$ or $T$ and with/without the logarithm.
- pg. 6: "Images include normal retinal tissue as well as [...] diabetic macular edema (DME), choroidal neovascularization (CNV), drusen and normal." - sentence seems not fully consistent.
- pg. 7: "All experiments are conducted on a single NVIDIA H100 GPU, although CPU inference is feasible due to the minimal number of trainable parameters." Since a frozen embedder is used, I would expect also CPU *training* to be possible.


[r1] Hu et al. Lora: Low-rank adaptation of large language models. ICLR, 1(2):3, 2022.
[r2] Zhou et al.: Learning to prompt for vision-language models. International Journal of Computer Vision, 130(9):2337–2348, 2022.

**Justification Of Final Rating:**

I would like to thank the authors for their efforts in addressing the points raised by the reviewers and especially for adding additional few-shot learning comparisons. The paper has merit in terms of the evaluation it presents; however, considerable concerns remain from my side.
1. In the revised version of the paper, the related work section still states that in-context learning for medical imaging [leaves] "classification, detection, and diagnostic reasoning largely unexplored", despite a significant number of papers looking into these tasks; additional references (to the ones mentioned in the original review) below [1, 2, 3, 4]. Co-op approaches, which make up a significant part of few-shot methods for medical image classification, are not mentioned.
2. An ablation study w.r.t. the learnable temperature parameter are missing (I again apologize for the late engagement in the discussion; however, the ablation study was mentioned under "weaknesses" not under "minor comments").
3. The paper evaluates PIKACHU on a highly simplified setting with a balanced support set on binary classification. This has been identified as a simplified and performance-wise overly-optimistic setting for clinical applications [2].

Taken together, this simplified view does not sufficiently address the clinical motivation described in the introduction, and - while extended experiments provide a better understanding of some related approaches - the paper does not fully address the current state-of-the-art for few-shot classification/ICL.

Since the other reviewers see this paper more favourably, I would like to strongly encourage this paper to tamper the concluding statements in the related work section and include additional works in a more differentiated manner in case of acceptance.

- [1] Taha Koleilat, Hojat Asgariandehkordi, Hassan Rivaz, Yiming Xiao: BiomedCoOp: Learning to Prompt for Biomedical Vision-Language Models, CVPR 2025, https://arxiv.org/abs/2411.15232
- [2] Julio Silva-Rodríguez, Fereshteh Shakeri, Houda Bahig, Jose Dolz, Ismail Ben Ayed: Few-Shot, Now for Real: Medical VLMs Adaptation without Balanced Sets or Validation, MICCAI 2025, https://arxiv.org/abs/2506.17500
- [3] Shakeri, Fereshteh, Huang, Yunshi, Silva-Rodriguez, Julio, Bahig, Houda, Tang, An, Dolz, Jose, Ben Ayed, Ismail:
Few-shot Adaptation of Medical Vision-Language Models, MICCAI 2024, https://doi.org/10.1007/978-3-031-72390-2_52
- [4] Shiri, Mahshid, Beyan, Cigdem, Murino, Vittorio: MadCLIP: Few-shot Medical Anomaly Detection with CLIP, MICCAI 2025, https://doi.org/10.1007/978-3-032-04978-0_40

**Justification Of The Preliminary Rating:**

The paper presents a rather broad evaluation of different foundation models on different tasks; however, there are concerns w.r.t. to the experiments and setup of the baselines, . While interesting and robust, the proposed "Prototypical In-context Knowledge Adaptation" is class mean classification / centroid classification, and with k-nn and linear probing a commonly used approach for using pretrained models. The classification tasks are simplified (binary classification) and still within "clean" datasets, from this reviewers perspective the challenges mentioned in the introduction are not sufficiently addressed to offset this missing methodological contribution with a strong use-case evaluation.

**Questions To Address In The Rebuttal:**

- Kindly clarify any misunderstandings w.r.t. to the method definition and the benefit of using a learned temperature parameter.
- Please more clearly describe the setup used for the baseline methods and why not other, similar few-shot learning approaches were included in the comparison.
- Please more clearly motivate the binary task - to what extent do you expect this to be representative for broader applicability?

---

> ### Author Response · Authors · 2026-01-25
>
> We thank the reviewer for their careful reading of the manuscript, constructive critique, and for highlighting both the strengths and limitations of our work. We also appreciate the positive feedback on the clarity of the presentation, the multi-dataset evaluation, and the inclusion of diverse foundation models.
>
> **[Clarification of misunderstandings w.r.t. to the method definition and benefit of using a learned temperature parameter]**: Clarifications regarding individual comments are provided in detail below including additional experiments. Using a learned temperature allows the model to control the scale of cosine similarity scores before the softmax operation, which directly affects class separation and prediction confidence. Without this scaling, similarities may be too compressed or overly peaked, leading to suboptimal gradients and unstable learning. By learning a temperature, the model adapts the sharpness of the decision boundaries to the task and backbone characteristics, improving gradient quality and convergence during episodic training.
>
> **[Experiments with other baselines and few-shot learning experiments]** Experiments with other few shot learning methods namely Tip-Adapter, Proto-Adapter and Lora  were added to the appendix C.1.  Results are shown below while addressing the detailed. These were not included originally because they require more trainable parameters compared to our method.
>
> **[Motivation for the binary task and representative for broader applicability]** This paper presents the first approach to use in-context learning with VLMs to make them generalizable on diverse medical image datasets. The paper initially had 72 experiments and we added 15 additional experiments to show various ablations associated with comparison to other methods and aggregation techniques. Extensions to other tasks are now possible. We therefore focus on balanced binary tasks to isolate the effects of the proposed similarity formulation and temperature scaling, without introducing additional confounders such as skewed class priors or dataset-specific heuristics.
>
> We now address the detailed comments provided by the reviewer:
>
> **[Updating the literature review to include other methods (e.g. LoRA)]** We have updated the literature review and as mentioned by other reviewers as well. An extensive comparison with other few/one-shot methods are also included in the appendix. This shows the effectiveness of our proposed method.
>
> **[Binary Classification]**: In this initial work, we present the first controlled and systematic study of in-context prototype construction with SOTA vision and vision-language foundation models. We therefore focus on balanced binary tasks to isolate the effects of the proposed similarity formulation and temperature scaling, without introducing additional confounders such as skewed class priors or dataset-specific heuristics.
>
> **[Predefined classifier weight]** The pre-trained weights refer to the original weights of the model for PubMedCLIP and SigLIP (available publicly). There is no prompt-based finetuning performed.
>
> **[Additional experiments on few shot methods such as Tip-Adapter, Proto-Adapter, Lora]** As mentioned by other reviewers, we have included additional comparisons to other few-shot based techniques in the appendix C.
> | Model | Strategy           | **ISIC** | **OCT** | **DR** |
> |------:|--------------------|:--------:|:-------:|:------:|
> |  | Baseline           | 0.49 | 0.50 | 0.50 |
> |  SigLIP      | ICL (KNN weighted) | 0.61 | 0.64 | 0.70 |
> |        | PIKACHU        | **0.73** | **0.83** | **0.77** |
>
> **[Results for SigLIP correctly listed?]** Yes, the numbers are correctly listed. Given that the test classes are balanced, we realize that reporting the AUC scores was not required and added some confusion. We updated Table 1 to now show only accuracies.
>
> **[prior text and pg. 10 Fig. 3 - I would have expected to see a performance range. Were multiple experiments with different samples performed?]** We now show results on multiple experiments and updated Figure 3 to include error bars.
>
> **Minor Comments**: Thanks to the reviewer for pointing out minor points (e.g. spelling) We have carefully reviewed the manuscript and made the changes throughout the paper. Some other minor comments and questions are addressed below:
>
> - **["If a class is absent in a particular episode (rare in balanced sampling), a zero vector is used by design."]** Correct. In fact, this shows the adaptability of our design and network architecture to multi-class settings.
>
>  - **["All experiments are conducted on a single NVIDIA H100 GPU, although CPU inference is feasible due to the minimal number of trainable parameters."]**  Yes, but please note that during inference, there are prototype computations that involve matrix multiplications. Thus, transferring 100% of computations to the CPU may add some additional time complexities.

---

> > ### Comment · Reviewer_WGZg · 2026-02-01
> >
> > Dear authors,
> > Thank you for the detailed response, I apologize for the late question.
> > - To the best of my understanding, the (revised) paper does not contain an ablation study w.r.t. the learning of the temperature parameter. Did I overlook this?
> > - What was the parameter K for the K-NN experiments?
> > - How is "strong calibration" defined in the context of this paper?
> > - "We now show results on multiple experiments and updated Figure 3 to include error bars." - where repeated experiments only performed for the impact of the support size K? How many? What differed between repetitions?

---

> > > ### Author Response · Authors · 2026-02-02
> > > **Additional Clarifications**
> > >
> > > We thank the reviewer for these detailed questions and the opportunity to clarify the experimental protocol and terminology used in the revised manuscript.
> > > 1. **Ablation on the learned temperature:**
> > >  Thank you for the comment. Since this request was raised under the minor comments, we did not include these results in the revised manuscript. Note that the current version already includes more than 15 additional experiments across multiple settings and datasets, which limited the scope for further ablations at this stage.
> > >
> > > 2. **Choice of K in the K-NN experiments:**
> > >  For all K-NN baselines, we used a fixed value of K=5. We will make this explicit in the experimental section to avoid any ambiguity.
> > >
> > > 3. **Definition of "strong calibration":**
> > >  In this work, we use the term strong calibration to emphasize that our method requires learning only a single scalar temperature parameter, yet achieves performance comparable to heavily parameterized adaptation methods such as LoRA or Proto-Adapter. In this sense, "strong" refers not to a formal probabilistic calibration definition, but to the strength of performance obtained under an extremely low-parameter regime. The corresponding quantitative comparisons are reported in the Appendix C.1.
> > >
> > > 4. **Repeated experiments and error bars in Figure 3:**
> > > For Figure 3, each point is obtained from six independent runs, using different random samplings of the support set and different initialization seeds, while keeping the backbone, prompts, and evaluation protocol fixed. We will explicitly report the number of repetitions and clearly describe the sources of randomness in the final version to improve clarity.

---

### Official Review · Reviewer_xvGf · 2026-01-15

**Confidence:** 3
**Preliminary Rating:** 3
**Final Rating:** 4

**Summary:**

This paper proposed the PIKACHU, a lightweight few-shot learning framework for medical imaging, designed to achieve rapid task adaptation without fine-tuning by freezing the base model and utilizing feature prototypes from the support set. This method only requires learning a single temperature coefficient parameter and uses cosine similarity for inference, thus avoiding catastrophic forgetting and reducing computational costs. Experiments show that this strategy achieves significant performance improvements compared to zero-shot baselines on heterogeneous datasets such as those for skin diseases, OCT, and diabetic retinopathy.

**Strengths:**

1. The article was validated on three vastly different medical datasets (dermoscopy, OCT, and fundus photography) and four different base models (visual-language models and purely visual models). This extensive experimental setup demonstrates its robustness as a general-purpose plugin.
2. By using minimized learnable parameters, the feasibility of deploying large models in local applications is improved.
3. The authors also verified the necessity of domain-specific pre-training by comparing with foundation model.

**Weaknesses:**

1. Innovation is limited, and there are some similar work before like (https://arxiv.org/pdf/1911.04623) . The author needs to conduct more literature research and compare different methods.
2. The baseline method used for comparison is not enough.
3. PIKACHU uses a simple "class mean" approach to construct prototypes (Equation 2 in the paper), which assumes that each disease class follows a unimodal Gaussian distribution in the feature space. However, medical imaging data often exhibits highly multimodal distributions. The proposed method may struggle to handle the multimodal distribution of medical data.

**Detailed Comments:**

None

**Justification Of Final Rating:**

I thank the authors for their detailed response and supplementary experiments, particularly the comparison with Tip-Adapter and LoRA. This effectively addresses my previous concerns regarding insufficient baselines and highlights PIKACHU's efficiency with only one learnable parameter. The explanation regarding the stability of distribution estimation in few-shot settings is reasonable, and the empirical results support the validity of using 'class-mean' prototypes in this context. However, the theoretical novelty remains incremental, and the evaluation is still primarily limited to balanced binary classification, with validation in complex multi-class scenarios needing further strengthening. Given that the method's practicality is well-supported and the major weaknesses have been addressed, I have decided to raise my score to 4.

**Justification Of The Preliminary Rating:**

My Borderline rating reflects a trade-off between the paper's strong practical utility and its limited methodological novelty. The extensive validation across three heterogeneous medical datasets and four foundation models is commendable, demonstrating the framework's robustness and feasibility for local deployment with minimal learnable parameters. However, the core innovation is incremental, as the proposed method is similar with established baseline (Simpleshot), utilizing nearly identical feature normalization and nearest-neighbor principles. Furthermore, the reliance on a simple "class mean" prototype implicitly assumes unimodal distributions, which may fail to capture the complex, multimodal nature inherent in diverse medical pathologies. Finally, the current evaluation relies too heavily on zero-shot baselines, and the authors should compare their approach against stronger, established few-shot learning methods to convincingly demonstrate superiority.

**Questions To Address In The Rebuttal:**

1. Tip-Adapter, Proto-Adapter, LoRA should be compared.
2. Compare with Simpleshot https://arxiv.org/pdf/1911.04623

---

> ### Author Response · Authors · 2026-01-25
>
> We thank the reviewer for their thoughtful evaluation and constructive feedback. We appreciate their recognition of the practical utility of the method, the extensive validation of the method across heterogeneous medical datasets, and the emphasis on minimal learnable parameters for deployability. Below, we address the raised concerns in detail.
>
> **1. Comparison to prior work**
> The reviewer requested to compare our work with SimpleShot (https://arxiv.org/abs/1911.04623#/ ). While SimpleShot focuses primarily on purely visual backbones (CNN), PIKACHU differs in several key aspects which are mentioned below (making a direct comparison to SimpleShot not straightforward).
>
>  - *Adaptation mechanism*: SimpleShot performs fully static nearest-neighbor or nearest-centroid classification in a fixed embedding space, whereas PIKACHU introduces an explicit in-context adaptation mechanism by learning a task-calibrating temperature parameter.
>  - *Role of the support set*: In SimpleShot, the support set is used only as reference points for distance computation; in contrast, PIKACHU treats the support set as in-context information that defines a task-specific prototype space for inference.
>  - *Backbone and scope*: SimpleShot is evaluated on CNN-based visual encoders and generic few-shot benchmarks, whereas PIKACHU leverages state-of-the art frozen foundation models (CLIP, SigLIP, PubMED-CLIP, Dinov2), including vision-language and medical FMs, under heterogeneous clinical distribution shifts. In SimpleShot, the training is done from scratch adding a time overhead while our proposed method uses foundation models as backbone and adapts to the support set.
> - *Objective*: SimpleShot serves as a strong metric-learning baseline demonstrating the importance of feature normalization, while PIKACHU is explicitly designed for test-time, in-context adaptation in low-data, real-world medical imaging scenarios. To reiterate, the goal of this paper was to introduce a method to adapt and evaluate a lightweight, in-context inference framework specifically tailored for classification using vision-language foundation models (VLFMs) and medical imaging scenarios.
>
>     That being said, in order to address the request made by the reviewer, we now show how a KNN performs when the features obtained from the frozen foundation models are used in a strategy similar to SimpleShot. These experiments and results are described in Appendix C.2.
>      | Model | Strategy           | **ISIC** | **OCT** | **DR** |
>      |------:|--------------------|:--------:|:-------:|:------:|
>      |  | Baseline           | 0.49 | 0.50 | 0.50 |
>      |  SigLIP      | ICL (KNN weighted) | 0.61 | 0.64 | 0.70 |
>      |        | PIKACHU        | **0.73** | **0.83** | **0.77** |
>
> 2. **Comparison with other few-shot baselines**
> We perform these additional experiments to compare a few other-shot adaptation methods, such as Tip-Adapter, Proto-Adapter, and LoRA with PIKACHU. These experiments and results are described in Appendix C.1.
> | Few-Shot Method     | ISIC | OCT | DR  | Trainable Params |
> |-----------|:----:|:---:|:---:|-----------------:|
> | Tip-Adapter | 0.71 | 0.78 | 0.74 | 1 |
> | Proto-Adapter | **0.73** | 0.82 | 0.76 | 99,137 |
> | LoRA (rank = 8) | 0.72 | **0.83** | 0.76 | 12,289 |
> | LoRA (rank = 16) | **0.73** | 0.82 | **0.77** | 24,577 |
> | PIKACHU  | **0.73** | **0.83** | **0.77** | **1** |
>
>     Our method consistently either outperforms the Tip-Adapter or is similar in performance to Proto-Adapter and Lora but with fewer trainable parameters.
>
> 3. **Prototype Construction and Multimodal Feature Distributions**
> The reviewer asked why we didn’t use and questioned our choice of mean-based prototypes. Our choice of mean-based prototypes was motivated by the fact that the extremely low-shot regime (often 1–10 examples), where estimating more complex distributions is statistically unstable (See Section 2.3 for a clarification). The reviewer correctly notes that using class-mean prototypes implicitly assumes unimodal feature distributions, and that medical imaging data can indeed exhibit significant intra-class heterogeneity and multimodality. However, despite the model’s seemingly simplified assumptions, large-scale experiments demonstrate a consistently strong performance across diverse medical imaging datasets and modalities, suggesting that foundation model (FM) embeddings already provide strong clustering that can be effectively leveraged even with simple aggregation. As such, a carefully designed in-context prototype framework can substantially improve few-shot medical image classification using frozen foundation models without episodic training or large memory overhead.

---

### Author Response · Authors · 2026-01-25

We thank the reviewers for their thorough and constructive feedback. All three reviewers provided a positive assessment of the manuscript, praising the clarity and practical impact of the proposed PIKACHU framework. They highlighted its simplicity and effectiveness in enabling rapid few-shot adaptation without fine-tuning, avoiding catastrophic forgetting, and requiring minimal learnable parameters, making it well suited for low-resource clinical settings. The reviewers also commended the clear and well-structured methodology, as well as the extensive experimental validation across diverse medical imaging modalities and backbone models, demonstrating the robustness and generality of the proposed approach.

Overall, there were a few overlapping concerns that we addressed in the revised manuscript:

 - **Comparison to other few-shot methods** [R1, R2]: We ran additional experiments with other few-shot methods such as Tip-Adapter, Proto-Adapter and Lora. The results are discussed in Appendix C.1 and PIKACHU either outperforms or achieves similar performance compared to these methods.
 - **Aggregation strategy such as KNN** [R1, R2, R3]: We ran additional experiment to test the performance of KNN-based aggregation. While this method achieves higher performance compared to the baseline, it is lower in performance compared to PIKACHU. This is discussed in details in Appendix C.2.
 - **Extension to multi-class classification** [R1, R2, R3]: We provide additional discussion and rationale for focusing on binary classification while addressing the reviewer’s comment individually.

Below, we respond to each reviewer’s specific comments and summarize the corresponding revisions in the updated manuscript. All modifications to the main paper and the appendix are highlighted in blue. Note that many of the newly added experiments and explanations are included in the appendix. Due to the nature of the proposed method, the paper initially included 72 experimental settings, with 15 additional experiments added in response to reviewers’ comments.

---

### Author Rebuttal · Authors · 2026-01-25

**Rebuttal:**

Please find our revised manuscript, including the appendix, where we have addressed all reviewers’ comments and concerns.

**Supporting Material:**

/attachment/9eebdbe220f80866281899d784a973f4a2135658.pdf

---

### Author Response · Authors · 2026-02-02
**Additional comments at the end of rebuttal**

We would like to sincerely thank all reviewers for their careful reading of our manuscript and for their constructive and insightful feedback. We believe that all major concerns, questions, and points of confusion raised by the reviewers have been fully clarified in this rebuttal. In addition, we have substantially strengthened the paper by including more than 15 new experiments and baselines across multiple datasets and settings.

---

### Author Response · Authors · 2026-02-02

Dear ACs and PCs,

We would like to note that Reviewer 2 requested additional experiments at a very late stage in the review process. Given the scope of these requests and the proximity to the rebuttal deadline, it was not feasible to conduct and include all of them within the available time. Nevertheless, we have made a substantial effort to strengthen the paper and have added multiple new experiments (15+) and baselines, as detailed in the rebuttal.

Warm Regards

---

### Meta-Review · Area_Chair_YiE2 · 2026-02-10

**Recommendation:** Accept (Poster)
**Confidence:** 4

**Metareview:**

This paper proposes PIKACHU, a very lightweight few-shot adaptation scheme for frozen vision(-language) foundation models in medical imaging, using class-mean prototypes in embedding space and learning only a single temperature parameter for cosine-similarity–based classification. Across three heterogeneous datasets (ISIC, OCT, DR) and four backbones (SigLIP, PubMedCLIP, DINOv2, ViT), the method consistently and substantially improves over zero-shot and linear-probe baselines, and the authors strengthen the rebuttal with additional comparisons to Tip-Adapter, Proto-Adapter, LoRA, and a KNN/SimpleShot-style baseline, generally matching or outperforming them with far fewer trainable parameters. Two reviewers (xvGf, 5chs) ultimately recommend weak accept, emphasizing the clarity, practicality, and broad experimental coverage, while noting that the method is conceptually simple and that evaluation is restricted to balanced binary classification on relatively clean datasets. A third reviewer (WGZg) remains at weak reject, arguing that the core idea reduces to well-known centroid classification, that the positioning of “in-context learning” and the related-work claims understate prior medical few-shot/ICL work, and that important aspects (temperature ablation, more realistic multiclass/imbalanced and clinically heterogeneous settings) are missing.

---

### Decision · Program_Chairs · 2026-02-13

Accept (Poster)